# Linear Convergence of a Frank-Wolfe Type Algorithm over Trace-Norm Balls[*]

**Zeyuan Allen-Zhu**
Microsoft Research, Redmond
zeyuan@csail.mit.edu

**Elad Hazan**
Princeton University
ehazan@cs.princeton.edu

**Wei Hu**
Princeton University
huwei@cs.princeton.edu

**Yuanzhi Li**
Princeton University
yuanzhil@cs.princeton.edu

## Abstract

We propose a rank-$k$ variant of the classical Frank-Wolfe algorithm to solve convex optimization over a trace-norm ball. Our algorithm replaces the top singular-vector computation (1-SVD) in Frank-Wolfe with a top-$k$ singular-vector computation ($k$-SVD), which can be done by repeatedly applying 1-SVD $k$ times. Alternatively, our algorithm can be viewed as a rank-$k$ restricted version of projected gradient descent. We show that our algorithm has a linear convergence rate when the objective function is smooth and strongly convex, and the optimal solution has rank at most $k$. This improves the convergence rate and the total time complexity of the Frank-Wolfe method and its variants.

## 1 Introduction

Minimizing a convex matrix function over a trace-norm ball, which is: (recall that the trace norm $\|X\|_*$ of a matrix $X$ equals the sum of its singular values)

$$\min_{X \in \mathbb{R}^{m \times n}} \left\{ f(X) \, : \, \|X\|_* \leq \theta \right\} \ , \tag{1.1}$$

is an important optimization problem that serves as a convex surrogate to many low-rank machine learning tasks, including matrix completion [2, 10, 16], multiclass classification [4], phase retrieval [3], polynomial neural nets [12], and more. In this paper we assume without loss of generality that $\theta = 1$.

One natural algorithm for Problem (1.1) is *projected gradient descent (PGD)*. In each iteration, PGD first moves $X$ in the direction of the gradient, and then projects it onto the trace-norm ball. Unfortunately, computing this projection requires the full singular value decomposition (SVD) of the matrix, which takes $O(mn \min\{m, n\})$ time in general. This prevents PGD from being efficiently applied to problems with large $m$ and $n$.

Alternatively, one can use *projection-free algorithms*. As first proposed by Frank and Wolfe [5], one can select a search direction (which is usually the gradient direction) and perform a linear optimization over the constraint set in this direction. In the case of Problem (1.1), performing linear optimization over a trace-norm ball amounts to computing the top (left and right) singular vectors of a matrix, which can be done much faster than full SVD. Therefore, projection-free algorithms become attractive for convex minimization over trace-norm balls.

Unfortunately, despite its low per-iteration complexity, the Frank-Wolfe (FW) algorithm suffers from slower convergence rate compared with PGD. When the objective $f(X)$ is smooth, FW requires $O(1/\varepsilon)$ iterations to convergence to an $\varepsilon$-approximate minimizer, and this $1/\varepsilon$ rate is tight even if the objective is also strongly convex [6]. In contrast, PGD achieves $1/\sqrt{\varepsilon}$ rate if $f(X)$ is smooth (under Nesterov's acceleration [14]), and $\log(1/\varepsilon)$ rate if $f(X)$ is both smooth and strongly convex.

---

[*]The full version of this paper can be found on https://arxiv.org/abs/1708.02105.

Recently, there were several results to revise the FW method to improve its convergence rate for strongly-convex functions. The $\log(1/\varepsilon)$ rate was obtained when the constraint set is a polyhedron [7, 11], and the $1/\sqrt{\varepsilon}$ rate was obtained when the constraint set is strongly convex [8] or is a spectrahedron [6].

Among these results, the spectrahedron constraint (i.e., for all positive semidefinite matrices $X$ with $\mathrm{Tr}(X) = 1$) studied by Garber [6] is almost identical to Problem (1.1), but slightly weaker.[2] When stating the result of Garber [6], we assume for simplicity that it also applies to Problem (1.1).

> **Our Question.** In this paper, we propose to study the following general question:
>
> Can we design a "rank-$k$ variant" of Frank-Wolfe to improve the convergence rate?
>
> (That is, in each iteration it computes the top $k$ singular vectors – i.e., $k$-SVD – of some matrix.)

Our motivation to study the above question can be summarized as follows:

- Since FW computes a 1-SVD and PGD computes a full SVD in each iteration, is there a value $k \ll \min\{n, m\}$ such that a rank-$k$ variant of FW can achieve the convergence rate of PGD?
- Since computing $k$-SVD costs roughly the same (sequential) time as "computing 1-SVD for $k$ times" (see recent work [1, 13]),[3] if using a rank-$k$ variant of FW, can the number of iterations be reduced by a factor more than $k$? If so, then we can improve the sequential running time of FW.
- $k$-SVD can be computed in a more *distributed* manner than 1-SVD. For instance, using block Krylov [13], one can distribute the computation of $k$-SVD to $k$ machines, each in charge of independent matrix-vector multiplications. Therefore, it is beneficial to study a rank-$k$ variant of FW in such settings.

## 1.1 Our Results

We propose `blockFW`, a rank-$k$ variant of Frank-Wolfe. Given a convex function $f(X)$ that is $\beta$-smooth, in each iteration $t$, `blockFW` performs an update $X_{t+1} \leftarrow X_t + \eta(V_t - X_t)$, where $\eta > 0$ is a constant step size and $V_t$ is a rank-$k$ matrix computed from the $k$-SVD of $(-\nabla f(X_t) + \beta\eta X_t)$. If $k = \min\{n, m\}$, `blockFW` can be shown to coincide with PGD, so it can also be viewed as a rank-$k$ restricted version of PGD.

**Convergence.** Suppose $f(X)$ is also $\alpha$-strongly convex and suppose the optimal solution $X^*$ of Problem (1.1) has rank $k$, then we show that `blockFW` achieves linear convergence: it finds an $\varepsilon$-approximate minimizer within $O(\frac{\beta}{\alpha} \log \frac{1}{\varepsilon})$ iterations, or equivalently, in

$$T = O\left(\frac{k\beta}{\alpha} \log \frac{1}{\varepsilon}\right) \quad \text{computations of 1-SVD.}$$

We denote by $T$ the number of 1-SVD computations throughout this paper. In contrast,

$$T_{\mathsf{FW}} = O\left(\frac{\beta}{\varepsilon}\right) \qquad\qquad\qquad\qquad\qquad \text{for Frank-Wolfe}$$

$$T_{\mathsf{Gar}} = O\left(\min\left\{\frac{\beta}{\varepsilon}, \quad \left(\frac{\beta}{\alpha}\right)^{1/4}\left(\frac{\beta}{\varepsilon}\right)^{3/4}\sqrt{k}, \quad \left(\frac{\beta}{\alpha}\right)^{1/2}\left(\frac{\beta}{\varepsilon}\right)^{1/2}\frac{1}{\sigma_{\min}(X^*)}\right\}\right) \quad \text{for Garber [6].}$$

Above, $\sigma_{\min}(X^*)$ is the minimum non-zero singular value of $X^*$. Note that $\sigma_{\min}(X^*) \le \frac{\|X^*\|_*}{\mathrm{rank}(X^*)} \le \frac{1}{k}$.

We note that $T_{\mathsf{Gar}}$ is always outperformed by $\min\{T, T_{\mathsf{FW}}\}$: ignoring the $\log(1/\varepsilon)$ factor, we have

- $\min\left\{\frac{\beta}{\varepsilon}, \frac{k\beta}{\alpha}\right\} \le \left(\frac{\beta}{\alpha}\right)^{1/4}\left(\frac{\beta}{\varepsilon}\right)^{3/4}k^{1/4} < \left(\frac{\beta}{\alpha}\right)^{1/4}\left(\frac{\beta}{\varepsilon}\right)^{3/4}\sqrt{k}$, and
- $\min\left\{\frac{\beta}{\varepsilon}, \frac{k\beta}{\alpha}\right\} \le \left(\frac{\beta}{\alpha}\right)^{1/2}\left(\frac{\beta}{\varepsilon}\right)^{1/2}k^{1/2} < \left(\frac{\beta}{\alpha}\right)^{1/2}\left(\frac{\beta}{\varepsilon}\right)^{1/2}\frac{1}{\sigma_{\min}(X^*)}$.

| algorithm | # rank | # iterations | time complexity per iteration |
|---|---|---|---|
| PGD [14] | $\min\{m,n\}$ | $\kappa \log(1/\varepsilon)$ | $O\big(mn \min\{m,n\}\big)$ |
| accelerated PGD [14] | $\min\{m,n\}$ | $\sqrt{\kappa} \log(1/\varepsilon)$ | $O\big(mn \min\{m,n\}\big)$ |
| Frank-Wolfe [9] | $1$ | $\frac{\beta}{\varepsilon}$ | $\tilde{O}\big(\mathtt{nnz}(\nabla)\big) \qquad \times \min\left\{ \frac{\|\nabla\|_2^{1/2}}{\varepsilon^{1/2}}, \frac{\|\nabla\|_2^{1/2}}{(\sigma_1(\nabla)-\sigma_2(\nabla))^{1/2}} \right\}$ |
| Garber [6] | $1$ | $\kappa^{\frac{1}{4}}\left(\frac{\beta}{\varepsilon}\right)^{\frac{3}{4}}\sqrt{k}$ , or $\kappa^{\frac{1}{2}}\left(\frac{\beta}{\varepsilon}\right)^{\frac{1}{2}}\frac{1}{\sigma_{\min}(X^*)}$ | $\tilde{O}\big(\mathtt{nnz}(\nabla)+(m+n)\big)$ $\times \min\left\{ \frac{\|\nabla\|_2^{1/2}}{\varepsilon^{1/2}}, \frac{\|\nabla\|_2^{1/2}}{(\sigma_1(\nabla)-\sigma_2(\nabla))^{1/2}} \right\}$ |
| `blockFW` | $k$ | $\kappa \log(1/\varepsilon)$ | $k \cdot \tilde{O}\big(\mathtt{nnz}(\nabla)+k(m+n)\kappa\big)$ $\times \min\left\{ \frac{(\|\nabla\|_2+\alpha)^{1/2}}{\varepsilon^{1/2}}, \frac{\kappa(\|\nabla\|_2+\alpha)^{1/2}}{\alpha^{1/2}\sigma_{\min}(X^*)} \right\}$ |

Table 1: Comparison of first-order methods to minimize a $\beta$-smooth, $\alpha$-strongly convex function over the unit-trace norm ball in $\mathbb{R}^{m \times n}$. In the table, $k$ is the rank of $X^*$, $\kappa = \frac{\beta}{\alpha}$ is the condition number, $\nabla = \nabla f(X_t)$ is the gradient matrix, $\mathtt{nnz}(\nabla)$ is the complexity to multiply $\nabla$ to a vector, $\sigma_i(X)$ is the $i$-th largest singular value of $X$, and $\sigma_{\min}(X)$ is the minimum non-zero singular value of $X$.

REMARK. The low-rank assumption on $X^*$ should be reasonable: as we mentioned, in most applications of Problem (1.1), the ultimate reason for imposing a trace-norm constraint is to ensure that the optimal solution is low-rank; otherwise the minimization problem may not be interesting to solve in the first place. Also, the immediate prior work [6] also assumes $X^*$ to have low rank.

**$k$-SVD Complexity.** For theoreticians who are concerned about the time complexity of $k$-SVD, we also compare it with the 1-SVD complexity of FW and Garber. If one uses `LazySVD` [1][4] to compute $k$-SVD in each iteration of `blockFW`, then the per-iteration $k$-SVD complexity can be bounded by

$$k \cdot \tilde{O}\big(\mathtt{nnz}(\nabla)+k(m+n)\kappa\big) \times \min\left\{ \frac{(\|\nabla\|_2+\alpha)^{1/2}}{\varepsilon^{1/2}}, \frac{\kappa(\|\nabla\|_2+\alpha)^{1/2}}{\alpha^{1/2}\sigma_{\min}(X^*)} \right\} \quad . \tag{1.2}$$

Above, $\kappa = \frac{\beta}{\alpha}$ is the condition number of $f$, $\nabla = \nabla f(X_t)$ is the gradient matrix of the current iteration $t$, $\mathtt{nnz}(\nabla)$ is the complexity to multiply $\nabla$ to a vector, $\sigma_{\min}(X^*)$ is the minimum non-zero singular value of $X^*$, and $\tilde{O}$ hides poly-logarithmic factors.

In contrast, if using Lanczos, the 1-SVD complexity for FW and Garber can be bounded as (see [6])

$$\tilde{O}\big(\mathtt{nnz}(\nabla)\big) \times \min\left\{ \frac{\|\nabla\|_2^{1/2}}{\varepsilon^{1/2}}, \frac{\|\nabla\|_2^{1/2}}{(\sigma_1(\nabla) - \sigma_2(\nabla))^{1/2}} \right\} \quad . \tag{1.3}$$

Above, $\sigma_1(\nabla)$ and $\sigma_2(\nabla)$ are the top two singular values of $\nabla$, and the gap $\sigma_1(\nabla) - \sigma_2(\nabla)$ can be as small as zero.

We emphasize that our $k$-SVD complexity (1.2) can be upper bounded by a quantity that only depends *poly-logarithmically* on $1/\varepsilon$. In contrast, the worst-case 1-SVD complexity (1.3) of FW and Garber depends on $\varepsilon^{-1/2}$ because the gap $\sigma_1 - \sigma_2$ can be as small as zero. Therefore, if one takes this additional $\varepsilon$ dependency into consideration for the convergence rate, then `blockFW` has rate $\mathrm{polylog}(1/\varepsilon)$, but FW and Garber have rates $\varepsilon^{-3/2}$ and $\varepsilon^{-1}$ respectively. The convergence rates and per-iteration running times of different algorithms for solving Problem (1.1) are summarized in Table 1.

**Practical Implementation.** Besides our theoretical results above, we also provide practical suggestions for implementing `blockFW`. Roughly speaking, one can automatically select a different "good" rank $k$ for each iteration. This can be done by iteratively finding the 1st, 2nd, 3rd, etc., top singular vectors of the underlying matrix, and then stop this process whenever the objective decrease is not worth further increasing the value $k$. We discuss the details in Section 6.

## 2 Preliminaries and Notation

For a positive integer $n$, we define $[n] := \{1, 2, \ldots, n\}$. For a matrix $A$, we denote by $\|A\|_F$, $\|A\|_2$ and $\|A\|_*$ respectively the Frobenius norm, the spectral norm, and the trace norm of $A$. We use $\langle \cdot, \cdot \rangle$ to denote the (Euclidean) inner products between vectors, or the (trace) inner products between matrices (i.e., $\langle A, B \rangle = \mathrm{Tr}(AB^\top)$). We denote by $\sigma_i(A)$ the $i$-th largest singular value of a matrix $A$, and by $\sigma_{\min}(A)$ the minimum *non-zero* singular value of $A$. We use $\mathrm{nnz}(A)$ to denote the time complexity of multiplying matrix $A$ to a vector (which is at most the number of non-zero entries of $A$). We define the *(unit) trace-norm ball* $\mathcal{B}_{m,n}$ in $\mathbb{R}^{m \times n}$ as $\mathcal{B}_{m,n} := \{X \in \mathbb{R}^{m \times n} : \|X\|_* \le 1\}$.

**Definition 2.1.** *For a differentiable convex function $f : \mathcal{K} \to \mathbb{R}$ over a convex set $\mathcal{K} \subseteq \mathbb{R}^{m \times n}$, we say*

- *$f$ is $\beta$-smooth if $f(Y) \le f(X) + \langle \nabla f(X), Y - X \rangle + \frac{\beta}{2}\|X - Y\|_F^2$ for all $X, Y \in \mathcal{K}$;*

- *$f$ is $\alpha$-strongly convex if $f(Y) \ge f(X) + \langle \nabla f(X), Y - X \rangle + \frac{\alpha}{2}\|X - Y\|_F^2$ for all $X, Y \in \mathcal{K}$.*

For Problem (1.1), we assume $f$ is differentiable, $\beta$-smooth, and $\alpha$-strongly convex over $\mathcal{B}_{m,n}$. We denote by $\kappa = \frac{\beta}{\alpha}$ the condition number of $f$, and by $X^*$ the minimizer of $f(X)$ over the trace-norm ball $\mathcal{B}_{m,n}$. The strong convexity of $f(X)$ implies:

**Fact 2.2.** $f(X) - f(X^*) \ge \frac{\alpha}{2}\|X - X^*\|_F^2$ *for all $X \in \mathcal{K}$.*

*Proof.* The minimality of $X^*$ implies $\langle \nabla f(X^*), X - X^* \rangle \ge 0$ for all $X \in \mathcal{K}$. The fact follows then from the $\alpha$-strong convexity of $f$. □

**The Frank-Wolfe Algorithm.** We now quickly review the Frank-Wolfe algorithm (see Algorithm 1) and its relation to PGD.

---

**Algorithm 1** Frank-Wolfe

---

**Input:** Step sizes $\{\eta_t\}_{t \ge 1}$ ($\eta_t \in [0, 1]$), starting point $X_1 \in \mathcal{B}_{m,n}$
 1: **for** $t = 1, 2, \ldots$ **do**
 2: $\quad V_t \leftarrow \mathrm{argmin}_{V \in \mathcal{B}_{m,n}} \langle \nabla f(X_t), V \rangle$
 $\qquad\qquad \diamond$ *by finding the top left/right singular vectors $u_t, v_t$ of $-\nabla f(X_t)$, and taking $V_t = u_t v_t^\top$.*
 3: $\quad X_{t+1} \leftarrow X_t + \eta_t(V_t - X_t)$
 4: **end for**

---

Let $h_t = f(X_t) - f(X^*)$ be the approximation error of $X_t$. The convergence analysis of Algorithm 1 is based on the following relation:

$$h_{t+1} = f(X_t + \eta_t(V_t - X_t)) - f(X^*) \overset{\text{①}}{\le} h_t + \eta_t \langle \nabla f(X_t), V_t - X_t \rangle + \frac{\beta}{2}\eta_t^2 \|V_t - X_t\|_F^2$$

$$\overset{\text{②}}{\le} h_t + \eta_t \langle \nabla f(X_t), X^* - X_t \rangle + \frac{\beta}{2}\eta_t^2 \|V_t - X_t\|_F^2 \overset{\text{③}}{\le} (1 - \eta_t)h_t + \frac{\beta}{2}\eta_t^2 \|V_t - X_t\|_F^2 .$$

$$(2.1)$$

Above, inequality ① uses the $\beta$-smoothness of $f$, inequality ② is due to the choice of $V_t$ in Line 2, and inequality ③ follows from the convexity of $f$. Based on (2.1), a suitable choice of the step size $\eta_t = \Theta(1/t)$ gives the convergence rate $O(\beta/\varepsilon)$ for the Frank-Wolfe algorithm.

If $f$ is also $\alpha$-strongly convex, a linear convergence rate can be achieved if we replace the linear optimization step (Line 2) in Algorithm 1 with a constrained quadratic minimization:

$$V_t \leftarrow \underset{V \in \mathcal{B}_{m,n}}{\mathrm{argmin}} \langle \nabla f(X_t), V - X_t \rangle + \frac{\beta}{2}\eta_t \|V - X_t\|_F^2 . \qquad (2.2)$$

In fact, if $V_t$ is defined as above, we have the following relation similar to (2.1):

$$h_{t+1} \le h_t + \eta_t \langle \nabla f(X_t), V_t - X_t \rangle + \frac{\beta}{2}\eta_t^2 \|V_t - X_t\|_F^2$$

$$\le h_t + \eta_t \langle \nabla f(X_t), X^* - X_t \rangle + \frac{\beta}{2}\eta_t^2 \|X^* - X_t\|_F^2 \le (1 - \eta_t + \kappa\eta_t^2)h_t ,$$

$$(2.3)$$

where the last inequality follows from Fact 2.2. Given (2.3), we can choose $\eta_t = \frac{1}{2\kappa}$ to obtain a linear convergence rate because $h_{t+1} \le (1 - 1/4\kappa)h_t$. This is the main idea behind the projected gradient

descent (PGD) method. Unfortunately, optimizing $V_t$ from (2.2) requires a projection operation onto $\mathcal{B}_{m,n}$, and this further requires a full singular value decomposition of the matrix $\nabla f(X_t) - \beta \eta_t X_t$.

# 3 A Rank-$k$ Variant of Frank-Wolfe

Our main idea comes from the following simple observation. Suppose we choose $\eta_t = \eta = \frac{1}{2\kappa}$ for all iterations, and suppose $\text{rank}(X^*) \le k$. Then we can add a low-rank constraint to $V_t$ in (2.2):

$$V_t \leftarrow \underset{V \in \mathcal{B}_{m,n}, \ \text{rank}(V) \le k}{\operatorname{argmin}} \langle \nabla f(X_t), V - X_t \rangle + \frac{\beta}{2} \eta \| V - X_t \|_F^2 \ . \tag{3.1}$$

Under this new choice of $V_t$, it is obvious that the same inequalities in (2.3) remain to hold, and thus the linear convergence rate of PGD can be preserved. Let us now discuss how to solve (3.1).

## 3.1 Solving the Low-Rank Quadratic Minimization (3.1)

Although (3.1) is non-convex, we prove that it can be solved efficiently. To achieve this, we first show that $V_t$ is in the span of the top $k$ singular vectors of $\beta \eta X_t - \nabla f(X_t)$.

**Lemma 3.1.** *The minimizer $V_t$ of (3.1) can be written as $V_t = \sum_{i=1}^{k} a_i u_i v_i^\top$, where $a_1, \ldots, a_k$ are nonnegative scalars, and $(u_i, v_i)$ is the pair of the left and right singular vectors of $A_t := \beta \eta X_t - \nabla f(X_t)$ corresponding to its $i$-th largest singular value.*

The proof of Lemma 3.1 is given in the full version of this paper. Now, owing to Lemma 3.1, we can perform $k$-SVD on $A_t$ to compute $\{(u_i, v_i)\}_{i \in [k]}$, plug the expression $V_t = \sum_{i=1}^{k} a_i u_i v_i^\top$ into the objective of (3.1), and then search for the optimal values $\{a_i\}_{i \in [k]}$. The last step is equivalent to minimizing $-\sum_{i=1}^{k} \sigma_i a_i + \frac{\beta}{2} \eta \sum_{i=1}^{k} a_i^2$ (where $\sigma_i = u_i^\top A_t v_i$) over the simplex $\Delta := \{a \in \mathbb{R}^k : a_1, \ldots, a_k \ge 0, \|a\|_1 \le 1\}$, which is the same as projecting the vector $\frac{1}{\beta \eta}(\sigma_1, \ldots, \sigma_k)$ onto the simplex $\Delta$. It can be easily solved in $O(k \log k)$ time (see for instance the applications in [15]).

## 3.2 Our Algorithm and Its Convergence

We summarize our algorithm in Algorithm 2 and call it `blockFW`.

---
**Algorithm 2** `blockFW`

---
**Input:** Rank parameter $k$, starting point $X_1 = 0$
1: $\eta \leftarrow \frac{1}{2\kappa}$.
2: **for** $t = 1, 2, \ldots$ **do**
3:     $A_t \leftarrow \beta \eta X_t - \nabla f(X_t)$
4:     $(u_1, v_1, \ldots, u_k, v_k) \leftarrow k\text{-SVD}(A_t)$
                                         $\diamond$ *$(u_i, v_i)$ is the $i$-th largest pair of left/right singular vectors of $A_t$*
5:     $a \leftarrow \operatorname{argmin}_{a \in \mathbb{R}^k, a \ge 0, \|a\|_1 \le 1} \|a - \frac{1}{\beta \eta}\sigma\|_2$                 $\diamond$ *where $\sigma := (u_i^\top A_t v_i)_{i=1}^{k}$*
6:     $V_t \leftarrow \sum_{i=1}^{k} a_i u_i v_i^\top$
7:     $X_{t+1} \leftarrow X_t + \eta(V_t - X_t)$
8: **end for**

---

Since the state-of-the-art algorithms for $k$-SVD are iterative methods, which in theory can only give approximate solutions, we now study the convergence of `blockFW` given *approximate $k$-SVD solvers*.

We introduce the following notion of an approximate solution to the low-rank quadratic minimization problem (3.1).

**Definition 3.2.** *Let $g_t(V) = \langle \nabla f(X_t), V - X_t \rangle + \frac{\beta}{2}\eta \|V - X_t\|_F^2$ be the objective function in (3.1), and let $g_t^* = g_t(X^*)$. Given parameters $\gamma \ge 0$ and $\varepsilon \ge 0$, a feasible solution $V$ to (3.1) is called $(\gamma, \varepsilon)$-approximate if it satisfies $g(V) \le (1 - \gamma)g_t^* + \varepsilon$.*

Note that the above multiplicative-additive definition makes sense because $g_t^* \le 0$:

**Fact 3.3.** *If $\text{rank}(X^*) \le k$, for our choice of step size $\eta = \frac{1}{2\kappa}$, we have $g_t^* = g_t(X^*) \le -(1 - \kappa\eta)h_t = -\frac{h_t}{2} \le 0$ according to (2.3).*

The next theorem gives the linear convergence of `blockFW` under the above approximate solutions to (3.1). Its proof is simple and uses a variant of (2.3) (see the full version of this paper).

**Theorem 3.4.** *Suppose* $\mathsf{rank}(X^*) \le k$ *and* $\varepsilon > 0$. *If each* $V_t$ *computed in* `blockFW` *is a* $(\frac{1}{2}, \frac{\varepsilon}{8})$-*approximate solution to (3.1), then for every $t$, the error $h_t = f(X_t) - f(X^*)$ satisfies*

$$h_t \le \left(1 - \tfrac{1}{8\kappa}\right)^{t-1} h_1 + \tfrac{\varepsilon}{2} \ .$$

*As a consequence, it takes $O(\kappa \log \frac{h_1}{\varepsilon})$ iterations to achieve the target error $h_t \le \varepsilon$.*

Based on Theorem 3.4, the per-iteration running time of `blockFW` is dominated by the time necessary to produce a $(\frac{1}{2}, \frac{\varepsilon}{8})$-approximate solution $V_t$ to (3.1), which we study in Section 4.

## 4 Per-Iteration Running Time Analysis

In this section, we study the running time necessary to produce a $(\frac{1}{2}, \varepsilon)$-approximate solution $V_t$ to (3.1). In particular, we wish to show a running time that depends only *poly-logarithmically* on $1/\varepsilon$. The reason is that, since we are concerning about the linear convergence rate (i.e., $\log(1/\varepsilon)$) in this paper, it is not meaningful to have a per-iteration complexity that scales polynomially with $1/\varepsilon$.

*Remark* 4.1. To the best of our knowledge, the Frank-Wolfe method and Garber's method [6] have their worst-case per-iteration complexities scaling polynomially with $1/\varepsilon$. In theory, this also slows down their overall performance in terms of the dependency on $1/\varepsilon$.

### 4.1 Step 1: The Necessary $k$-SVD Accuracy

We first show that if the $k$-SVD in Line 4 of `blockFW` is solved sufficiently accurate, then $V_t$ obtained in Line 6 will be a sufficiently good approximate solution to (3.1). For notational simplicity, in this section we denote $G_t := \|\nabla f(X_t)\|_2 + \alpha$, and we let $k^* = \mathsf{rank}(X^*) \le k$.

**Lemma 4.2.** *Suppose $\gamma \in [0, 1]$ and $\varepsilon \ge 0$. In each iteration $t$ of* `blockFW`*, if the vectors $u_1, v_1, \ldots, u_k, v_k$ returned by $k$-SVD in Line 4 satisfy $u_i^\top A_t v_i \ge (1 - \gamma)\sigma_i(A_t) - \varepsilon$ for all $i \in [k^*]$, then $V_t = \sum_{i=1}^k a_i u_i v_i^\top$ obtained in Line 6 is $\left(\left(\frac{6G_t}{h_t} + 2\right)\gamma, \varepsilon\right)$-approximate to (3.1).*

The proof of Lemma 4.2 is given in the full version of this paper, and is based on our earlier characterization Lemma 3.1.

### 4.2 Step 2: The Time Complexity of $k$-SVD

We recall the following complexity statement for $k$-SVD:

**Theorem 4.3** ([1]). *The running time to compute the $k$-SVD of $A \in \mathbb{R}^{m \times n}$ using* `LazySVD` *is*[5]

$$\tilde{O}\left(\frac{k \cdot \mathsf{nnz}(A) + k^2(m+n)}{\sqrt{\gamma}}\right) \quad or \quad \tilde{O}\left(\frac{k \cdot \mathsf{nnz}(A) + k^2(m+n)}{\sqrt{\mathsf{gap}}}\right) \ .$$

*In the former case, we can have $u_i^\top A v_i \ge (1 - \gamma)\sigma_i(A)$ for all $i \in [k]$; in the latter case, if $\mathsf{gap} \in \left(0, \frac{\sigma_{k^*}(A) - \sigma_{k^*+1}(A)}{\sigma_{k^*}(A)}\right]$ for some $k^* \in [k]$, then we can guarantee $u_i^\top A v_i \ge \sigma_i(A) - \varepsilon$ for all $i \in [k^*]$.*

**The First Attempt.** Recall that we need a $(\frac{1}{2}, \varepsilon)$-approximate solution to (3.1). Using Lemma 4.2, it suffices to obtain a $(1 - \gamma)$-multiplicative approximation to the $k$-SVD of $A_t$ (i.e., $u_i^\top A_t v_i \ge (1 - \gamma)\sigma_i(A_t)$ for all $i \in [k]$), as long as $\gamma \le \frac{1}{12G_t/h_t+4}$. Therefore, we can directly apply the *first* running time in Theorem 4.3: $\tilde{O}\left(\frac{k \cdot \mathsf{nnz}(A_t) + k^2(m+n)}{\sqrt{\gamma}}\right)$. However, when $h_t$ is very small, this running time can be unbounded. In that case, we observe that $\gamma = \frac{\varepsilon}{G_t}$ (independent of $h_t$) also suffices: since $\|A_t\|_2 = \left\|\frac{\alpha}{2} X_t - \nabla f(X_t)\right\|_2 \le \frac{\alpha}{2} + \|\nabla f(X_t)\|_2 \le G_t$, from $u_i^\top A_t v_i \ge (1 - \varepsilon/G_t)\sigma_i(A_t)$ we have $u_i^\top A_t v_i \ge \sigma_i(A_t) - \frac{\varepsilon}{G_t}\sigma_i(A_t) \ge \sigma_i(A_t) - \frac{\varepsilon}{G_t}\|A_t\|_2 \ge \sigma_i(A_t) - \varepsilon$; then according to Lemma 4.2 we can obtain $(0, \varepsilon)$-approximation to (3.1), which is stronger than $(\frac{1}{2}, \varepsilon)$-approximation. We summarize this running time (using $\gamma = \frac{\varepsilon}{G_t}$) in Claim 4.5; the running time depends *polynomially* on $\frac{1}{\varepsilon}$.

**The Second Attempt.** To make our linear convergence rate (i.e., the $\log(1/\varepsilon)$ rate) meaningful, we want the $k$-SVD running time to depend *poly-logarithmically* on $1/\varepsilon$. Therefore, when $h_t$ is small, we wish to instead apply the *second* running time in Theorem 4.3.

Recall that $X^*$ has rank $k^*$ so $\sigma_{k^*}(X^*) - \sigma_{k^*+1}(X^*) = \sigma_{\min}(X^*)$. We can show that this implies $A^* := \frac{\alpha}{2}X^* - \nabla f(X^*)$ also has a large gap $\sigma_{k^*}(A^*) - \sigma_{k^*+1}(A^*)$. Now, according to Fact 2.2, when $h_t$ is small, $X_t$ and $X^*$ are sufficiently close. This means $A_t = \frac{\alpha}{2}X_t - \nabla f(X_t)$ is also close to $A^*$, and thus has a large gap $\sigma_{k^*}(A_t) - \sigma_{k^*+1}(A_t)$. Then we can apply the second running time in Theorem 4.3.

### 4.2.1 Formal Running Time Statements

**Fact 4.4.** *We can store $X_t$ as a decomposition into at most $\mathsf{rank}(X_t) \leq kt$ rank-1 components.[6] Therefore, for $A_t = \frac{\alpha}{2}X_t - \nabla f(X_t)$, we have $\mathsf{nnz}(A_t) \leq \mathsf{nnz}(\nabla f(X_t)) + (m+n)\mathsf{rank}(X_t) \leq \mathsf{nnz}(\nabla f(X_t)) + (m+n)kt$.*

If we always use the first running time in Theorem 4.3, then Fact 4.4 implies:

**Claim 4.5.** *The $k$-SVD computation in the $t$-th iteration of* `blockFW` *can be implemented in $\tilde{O}\big((k \cdot \mathsf{nnz}(\nabla f(X_t)) + k^2(m+n)t)\sqrt{G_t/\varepsilon}\big)$ time.*

*Remark* 4.6. As long as $(m+n)kt \leq \mathsf{nnz}(\nabla f(X_t))$, the $k$-SVD running time in Claim 4.5 becomes $\tilde{O}\big(k \cdot \mathsf{nnz}(\nabla f(X_t))\sqrt{G_t/\varepsilon}\big)$, which roughly equals $k$-times the 1-SVD running time $\tilde{O}\big(\mathsf{nnz}(\nabla)\sqrt{\|\nabla\|_2/\varepsilon}\big)$ of FW and Garber [6]. Since in practice, it suffices to run `blockFW` and FW for a few hundred 1-SVD computations, the relation $(m+n)kt \leq \mathsf{nnz}(\nabla f(X_t))$ is often satisfied.

If, as discussed above, we apply the first running time in Theorem 4.3 only for large $h_t$, and apply the second running time in Theorem 4.3 for small $h_t$, then we obtain the following theorem whose proof is given in the full version of this paper.

**Theorem 4.7.** *The $k$-SVD comuputation in the $t$-th iteration of* `blockFW` *can be implemented in $\tilde{O}\Big(\big(k \cdot \mathsf{nnz}(\nabla f(X_t)) + k^2(m+n)t\big)\frac{\kappa\sqrt{G_t/\alpha}}{\sigma_{\min}(X^*)}\Big)$ time.*

*Remark* 4.8. Since according to Theorem 3.4 we only need to run `blockFW` for $O(\kappa \log(1/\varepsilon))$ iterations, we can plug $t = O(\kappa \log(1/\varepsilon))$ into Claim 4.5 and Theorem 4.7, and obtain the running time presented in (1.2). The per-iteration running time of `blockFW` depends *poly-logarithmically* on $1/\varepsilon$. In contrast, the per-iteration running times of Garber [6] and FW depend *polynomially* on $1/\varepsilon$, making their total running times even worse in terms of dependency on $1/\varepsilon$.

## 5 Maintaining Low-Rank Iterates

One of the main reasons to impose trace-norm constraints is to produce low-rank solutions. However, the rank of iterate $X_t$ in our algorithm `blockFW` can be as large as $kt$, which is much larger than $k$, the rank of the optimal solution $X^*$. In this section, we show that by adding a simple modification to `blockFW`, we can make sure the rank of $X_t$ is $O(k\kappa \log \kappa)$ in all iterations $t$, without hurting the convergence rate much.

We modify `blockFW` as follows. Whenever $t-1$ is a multiple of $S = \lceil 8\kappa(\log\kappa + 1)\rceil$, we compute (note that this is the same as setting $\eta = 1$ in (3.1))

$$W_t \leftarrow \underset{W \in \mathcal{B}_{m,n},\ \mathsf{rank}(W) \leq k}{\operatorname{argmin}} \langle \nabla f(X_t), W - X_t\rangle + \frac{\beta}{2}\|W - X_t\|_F^2 \ ,$$

and let the next iterate $X_{t+1}$ be $W_t$. In all other iterations the algorithm is unchanged. After this change, the function value $f(X_{t+1})$ may be greater than $f(X_t)$, but can be bounded as follows:

**Lemma 5.1.** *Suppose $\mathsf{rank}(X^*) \leq k$. Then we have $f(W_t) - f(X^*) \leq \kappa h_t$.*

*Proof.* We have the following relation similar to (2.3):

$$f(W_t) - f(X^*) \leq h_t + \langle \nabla f(X_t), W_t - X_t\rangle + \frac{\beta}{2}\|W_t - X_t\|_F^2$$

$$\leq h_t + \langle \nabla f(X_t), X^* - X_t\rangle + \frac{\beta}{2}\|X^* - X_t\|_F^2$$

$$\leq h_t - h_t + \frac{\beta}{2} \cdot \frac{2}{\alpha}h_t \quad = \kappa h_t \ . \qquad \square$$

From Theorem 3.4 we know that $h_{S+1} \leq (1 - \frac{1}{8\kappa})^S h_1 + \frac{\varepsilon}{2} \leq (1 - \frac{1}{8\kappa})^{8\kappa(\log \kappa + 1)} h_1 + \frac{\varepsilon}{2} \leq e^{-(\log \kappa + 1)} h_1 + \frac{\varepsilon}{2} = \frac{1}{e\kappa} h_1 + \varepsilon/2$. Therefore, after setting $X_{S+2} = W_{S+1}$, we still have $h_{S+2} \leq \frac{1}{e} h_1 + \frac{\kappa \varepsilon}{2}$ (according to Lemma 5.1). Continuing this analysis (letting the $\kappa \varepsilon$ here be the "new $\varepsilon$"), we know that this modified version of blockFW converges to an $\varepsilon$-approximate minimizer in $O\left(\kappa \log \kappa \cdot \log \frac{h_1}{\varepsilon}\right)$ iterations.

*Remark* 5.2. Since in each iteration the rank of $X_t$ is increased by at most $k$, if we do the modified step every $S = O(\kappa \log \kappa)$ iterations, we have that throughout the algorithm, $\mathsf{rank}(X_t)$ is never more than $O(k\kappa \log \kappa)$. Furthermore we can always store $X_t$ using $O(k\kappa \log \kappa)$ vectors, instead of storing all the singular vectors obtained in previous iterations.

# 6 Preliminary Empirical Evaluation

We conclude this paper with some preliminary experiments to test the performance of blockFW. We first recall two machine learning tasks that fall into Problem (1.1).

**Matrix Completion.** Suppose there is an unknown matrix $M \in \mathbb{R}^{m \times n}$ close to low-rank, and we observe a subset $\Omega$ of its entries – that is, we observe $M_{i,j}$ for every $(i,j) \in \Omega$. (Think of $M_{i,j}$ as user $i$'s rating of movie $j$.) One can recover $M$ by solving the following convex program:

$$\min_{X \in \mathbb{R}^{m \times n}} \left\{ \frac{1}{2} \sum_{(i,j) \in \Omega} (X_{i,j} - M_{i,j})^2 \mid \|X\|_* \leq \theta \right\} . \tag{6.1}$$

Although Problem (6.1) is not strongly convex, our experiments show the effectiveness of blockFW on this problem.

**Polynomial Neural Networks.** Polynomial networks are neural networks with quadratic activation function $\sigma(a) = a^2$. Livni et al. [12] showed that such networks can express any function computed by a Turing machine, similar to networks with ReLU or sigmoid activations. Following [12], we consider the class of 2-layer polynomial networks with inputs from $\mathbb{R}^d$ and $k$ hidden neurons:

$$P_k = \left\{ x \mapsto \sum_{j=1}^{k} a_j (w_j^\top x)^2 \,\middle|\, \forall j \in [k], w_j \in \mathbb{R}^d, \|w_j\|_2 = 1 \bigwedge a \in \mathbb{R}^k \right\} .$$

If we write $A = \sum_{i=1}^{k} a_j w_j w_j^\top$, we have the following equivalent formulation:

$$P_k = \left\{ x \mapsto x^\top A x \,\middle|\, A \in \mathbb{R}^{d \times d}, \mathsf{rank}(A) \leq k \right\} .$$

Therefore, if replace the hard rank constraint with trace norm $\|A\|_* \leq \theta$, the task of empirical risk minimization (ERM) given training data $\{(x_1, y_1), \ldots, (x_N, y_N)\} \subset \mathbb{R}^d \times \mathbb{R}$ can be formulated as[7]

$$\min_{A \in \mathbb{R}^{d \times d}} \left\{ \frac{1}{2} \sum_{i=1}^{N} (x_i^\top A x_i - y_i)^2 \mid \|A\|_* \leq \theta \right\} . \tag{6.2}$$

Since $f(A) = \frac{1}{2} \sum_{i=1}^{N} (x_i^\top A x_i - y_i)^2$ is convex in $A$, the above problem falls into Problem (1.1). Again, this objective $f(A)$ might not be strongly convex, but we still perform experiments on it.

## 6.1 Preliminary Evaluation 1: Matrix Completion on Synthetic Data

We consider the following synthetic experiment for matrix completion. We generate a random rank-10 matrix in dimension $1000 \times 1000$, plus some small noise. We include each entry into $\Omega$ with probability $1/2$. We scale $M$ to $\|M\|_* = 10000$, so we set $\theta = 10000$ in (6.1).

We compare blockFW with FW and Garber [6]. When implementing the three algorithms, we use exact line search. For Garber's algorithm, we tune its parameter $\eta_t = \frac{c}{t}$ with different constant values $c$, and then exactly search for the optimum $\tilde{\eta}_t$. When implementing blockFW, we use $k = 10$ and $\eta = 0.2$. We use the MATLAB built-in solver for 1-SVD and $k$-SVD.

In Figure 1(a), we compare the numbers of 1-SVD computations for the three algorithms. The plot confirms that it suffices to apply a rank-$k$ variant FW in order to achieve linear convergence.

## 6.2 Auto Selection of $k$

In practice, it is often unrealistic to know $k$ in advance. Although one can simultaneously try $k = 1, 2, 4, 8, \ldots$ and output the best possible solution, this can be unpleasant to work with. We propose the following modification to blockFW which automatically chooses $k$.

In each iteration $t$, we first run 1-SVD and compute the objective decrease, denoted by $d_1 \geq 0$. Now, given any approximate $k$-SVD decomposition of the matrix $A_t = \beta \eta X_t - \nabla f(X_t)$, we can compute its $(k + 1)$-SVD using one additional 1-SVD computation according to the LazySVD framework [1].

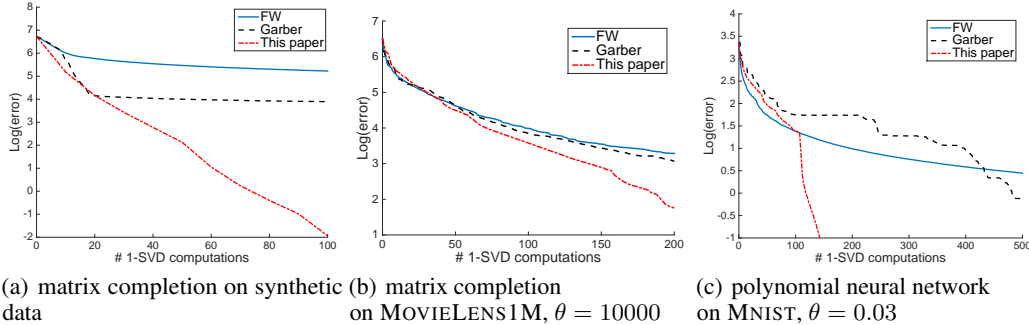

(a) matrix completion on synthetic data

(b) matrix completion on MOVIELENS1M, $\theta = 10000$

(c) polynomial neural network on MNIST, $\theta = 0.03$

Figure 1: Partial experimental results. The full 6 plots for MOVIELENS and 3 plots for MNIST are included in the full version of this paper.

We compute the new objective decrease $d_{k+1}$. We stop this process and move to the next iteration $t+1$ whenever $\frac{d_{k+1}}{k+1} < \frac{d_k}{k}$. In other words, we stop whenever it "appears" not worth further increasing $k$. We count this iteration $t$ as using $k + 1$ computations of 1-SVD.

All the experiments on real-life datasets are performed using this above auto-$k$ process.

### 6.3 Preliminary Evaluation 2: Matrix Completion on MOVIELENS

We study the same experiment in Garber [6], the matrix completion Problem (6.1) on datasets MOVIELENS100K ($m = 943, n = 1862$ and $|\Omega| = 10^5$) and MOVIELENS1M ($m = 6040, n = 3952$ and $|\Omega| \approx 10^6$). In the second dataset, following [6], we further subsample $\Omega$ so it contains about half of the original entries. For each dataset, we run FW, Garber, and `blockFW` with three different choices of $\theta$.[8] We present the six plots side-by-side in the full version of this paper.

We observe that when $\theta$ is large, there is no significant advantage for using `blockFW`. This is because the rank of the optimal solution $X^*$ is also high for large $\theta$. In contrast, when $\theta$ is small (so $X^*$ is of low rank), as demonstrated for instance by Figure 1(b), it is indeed beneficial to apply `blockFW`.

### 6.4 Preliminary Evaluation 3: Polynomial Neural Network on MNIST

We use the 2-layer neural network Problem (6.2) to train a binary classifier on the MNIST dataset of handwritten digits, where the goal is to distinguish images of digit "0" from images of other digits. The training set contains $N = 60000$ examples each of dimension $d = 28 \times 28 = 784$. We set $y_i = 1$ if that example belongs to digit "0" and $y_i = 0$ otherwise. We divide the original grey levels by 256 so $x_i \in [0, 1]^d$. We again try three different values of $\theta$, and compare FW, Garber, and `blockFW`.[9] We present the three plots side-by-side in the full version of this paper.

The performance of our algorithm is comparable to FW and Garber for large $\theta$, but as demonstrated for instance by Figure 1(c), when $\theta$ is small so $\mathrm{rank}(X^*)$ is small, it is beneficial to use `blockFW`.

## 7 Conclusion

In this paper, we develop a rank-$k$ variant of Frank-Wolfe for Problem (1.1) and show that: (1) it converges in $\log(1/\varepsilon)$ rate for smooth and strongly convex functions, and (2) its per-iteration complexity scales with $\mathrm{polylog}(1/\varepsilon)$. Preliminary experiments suggest that the value $k$ can also be automatically selected, and our algorithm outperforms FW and Garber [6] when $X^*$ is of relatively smaller rank.

We hope more rank-$k$ variants of Frank-Wolfe can be developed in the future.

### Acknowledgments

Elad Hazan was supported by NSF grant 1523815 and a Google research award. The authors would like to thank Dan Garber for sharing his code for [6].

## Footnotes

[2]The the best of our knowledge, given an algorithm that works for spectrahedron, to solve Problem (1.1), one has to define a function $g(Y)$ over $(n + m) \times (n + m)$ matrices, by setting $g(Y) = f(2Y_{1:m, m+1:m+n})$ [10]. After this transformation, the function $g(Y)$ is no longer strongly convex, even if $f(X)$ is strongly convex. In contrast, most algorithms for trace-norm balls, including FW and our later proposed algorithm, work as well for spectrahedron after minor changes to the analysis.

[3]Using block Krylov [13], Lanszos [1], or SVRG [1], at least when $k$ is small, the time complexity of (approximately) computing the top $k$ singular vectors of a matrix is no more than $k$ times the complexity of (approximately) computing the top singular vector of the same matrix. We refer interested readers to [1] for details.

[4]In fact, `LazySVD` is a general framework that says, with a meaningful theoretical support, one can apply a reasonable 1-SVD algorithm $k$ times in order to compute $k$-SVD. For simplicity, in this paper, whenever referring to `LazySVD`, we mean to apply the Lanczos method $k$ times.

[5]The first is known as the *gap-free* result because it does not depend on the gap between any two singular values. The second is known as the *gap-dependent* result, and it requires a $k \times k$ full SVD after the $k$ approximate singular vectors are computed one by one. The $\tilde{O}$ notation hides poly-log factors in $1/\varepsilon$, $1/\gamma$, $m$, $n$, and $1/\mathsf{gap}$.

[6] In Section 5, we show how to ensure that $\mathsf{rank}(X_t)$ is always $O(k\kappa \log \kappa)$, a quantity independent of $t$.

[7]We consider square loss for simplicity. It can be any loss function $\ell(x_i^\top A x_i, y_i)$ convex in its first argument.

[8]We perform exact line search for all algorithms. For Garber [6], we tune the best $\eta_t = \frac{c}{t}$ and exactly search for the optimal $\tilde{\eta}_t$. For `blockFW`, we let $k$ be chosen automatically and choose $\eta = 0.01$ for all the six experiments.

[9]We perform exact line search for all algorithms. For Garber [6], we tune the best $\eta_t = \frac{c}{t}$ and exactly search for the optimal $\tilde{\eta}_t$. For `blockFW`, we let $k$ be chosen automatically and choose $\eta = 0.0005$ for all the three experiments.

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
