[Reviews · NeurIPS 2017]

Reviewer 1



The result considers a very relevant and realistic setting when the true solution has low rank, and the algorithm is very simple. The paper is creative, overall well written and most times clearly distinguishes the presented results from existing ones. The additional practical impact compared to [6] might not be huge but still valuable, and the paper gives several interesting insights on the side. A unfortunate point at the moment is that the algorithm is not set into context with projected gradient onto a rank-k constraint (with bounded nuclear norm, and importantly in the otherwise unconstrained case also). Your Equation (3.1) is less closer to FW but much more to projected gradient, for which a rich literature already exists. This is especially blatant for the practical rank-k iterates version (Section 5), where the new iterate simply is W_t (as in PG), and no FW step update is even performed. Could your result be extended to k-restricted strong convexity (only strongly convex over the rank-k directions). Plenty of papers have studied analogous results for the vector case, but are not cited here. Equation (1.2) and following: You don't clearly explain the dependence on T for your method. The discussion of eps-dependence on the inner loop complexity of [6] is appreciated, but you don't explain why lazySVD can't be used within [6] as well. At least explain how your Theorem 4.3 with k=1 serves the rank-1 case. Also, the paper would benefit from a more clear discussion of the approximation quality to which the internal (1,k)-SVD need to be solved (which is well known for the FW case). It is discussed nicely for the new method, but not for the rank-1 ones (FW and [6]). Instead of some of the more technical lemmas and details, I'd prefer to have more on Section 5 (keeping the iterates rank k) in the paper, as this is more crucial for potential users. Maybe add the word 'iterates' in the title for clarity. - minimum non-zero singular value of X^*, so it satisfies \sigma_min(X^*) <= 1/k: "so that it satisfies" should be rephrased: If it is satisfied, your comparison holds, if not, it is not outperformed, right? Minor comments: - line 29: Recently, there were - second footnote on page 2: "with a meaningful theoretical support", "reasonable": what is meant here? better reformulate - line 218: Turing completeness: for how many layers? == update after rebuttal == Please clarify the nomenculature and equivalence with PGD with projection onto the non-convex set of the rank-constraint, as this is crucial for the context of the paper. Otherwise, the authors have clarified most questions. For restricted strong convexity, indeed please discuss at least the direction and concept. See e.g. https://arxiv.org/pdf/1703.02721 for the matrix case, and for the easier vector case e.g. http://papers.nips.cc/paper/4412-greedy-algorithms-for-structurally-constrained-high-dimensional-problems.pdf

Reviewer 2



---- After the rebuttal ---- I cannot help but notice some similarities of this method and the CLASH method in the vector case: https://arxiv.org/pdf/1203.2936.pdf The CLASH authors in effect make a projection onto the rank-k ball (corresponding to Combinatorial part in the CLASH method), followed by the projection onto the rank constraint (Least Absolute SHrinkage part). The latter is now a vector problem thanks to the affine invariance of the Frobenius norm in this paper. The authors in the CLASH work on the vector case directly; however, as opposed to performing the simple shrinkage update as in this paper, the CLASH authors actually solve the "fully corrective problem." Perhaps, this would be of interest to the authors. ----- The main idea of this paper is based on a simple observation: injecting a rank-k constraint into the subproblem in PGD does not affect the convergence rate, as soon as the solution rank is less than or equal to k. This simple observation however leads to an algorithm with linear convergence rate that possess the benefits of FW method: low per iteration computational cost and low-rank updates. Note however, from a purely mathematical standpoint, I would argue calling this method as a Frank-Wolfe variant, as the rigorous description of FW is composed of the computation of a search direction and the application of a linear minimization oracle, followed with an averaging step. This algorithm however is based on a nonconvex quadratic subproblem, but this subproblem can be solved with theoretical guarantees efficiently. That would be interesting to see the practical implications of the results section 5 that bounds the rank of the decision variable. Did you try this variant with replacement step in some numerical experiments? I would suggest addition of these results into the appendix. Note that there is a particular recent research direction about the storage efficiency of FW variants, e.g. [1]. In the current version of the manuscript, theoretical guarantees hold only if the solution rank is less than or equal to the input rank k. In practice, however, the solution is not exactly low rank due to the presence of the noise and imperfections in tuning the constraint radius. It is approximately low rank with a fast decay in the spectrum. As far as I can see the proof still holds when solution X* is replaced with the best rank-k approximation of the solution [X*]_k (see e.g. the guarantees provided in [1]). This would impose theoretical guarantees when f(X*) – f([X*]_k) is small. The numerical experiments fall short of satisfaction for the nips venue. All the experiments with real data are solved using the auto selection of k heuristic. It is not clear from the text if this heuristic still possesses the theoretical guarantees of the original variant where k is fixed in advance. Plots with #1-SVD computations at x axis are very useful to compare the performance in terms of the overall computational cost. However, I would still suggest adding some information about the k chosen at each iteration while solving the real data problems. It would be also interesting to see some practical evidence of what happens when the input rank k is chosen slightly less than the solution rank, or when the solution is not low rank but it has a fast decay in the singular value spectrum. As noted in the paper, algorithm outperforms other convex competitors in the convergence rate when the constraint radius is small, but there is no significant advantage when the radius is large. This is an expected result as the smaller radius tempts a lower rank solution. However, it is not possible to deduct which values of the radius are practically reasonable in the experiments presented in the paper (i.e. which values of theta results in lower test error). [1] Yurtsever et.al. Sketchy decisions: Convex low-rank matrix optimization with optimal storage. AISTATS, 2017.

Reviewer 3



The paper modified the Frank-Wolfe algorithm for strongly convex optimization over the trace-norm ball, by changing 1-svd with k-svd. It is proven that the convergence rate was improved from sublinear to superlinear. The theoretical results are solid and thorough. Experimental results show advantages of the proposed algorithm. I have the following minor comments: 1. It is understandable that if the rank is underestimated, then the optimal solution will not be approached. However, it is unclear whether if the rank is overestimated, then the optimal solution can still be achieved. Such an issue should be analyzed or evaluaed. 2. I don't know why the minimum nonzero singular value of X^* should be <= 1/k in line 51. 3. Although the theorems assumed strong convexity, the examplar models in experiments are not strongly convex. Can the authors replace them with other practical problems that fit for the assumptions? 4. In polynomial neural network, I believe that the rank cannot be small for good classification performance. Finally, the authors revealed their identity by writing "our immediate prior work [6]" in line 58. === review update == I don't object accepting the paper if the identity-revealing issue is ignored. Hope what the authors said "none of the author(s) of the current paper is an author of [6]." is true.